# Fostering Youth Female Athletes’ Decision-Making Skills through Competitive Volleyball: A Mixed Methods Design

**DOI:** 10.3390/ijerph192013261

**Published:** 2022-10-14

**Authors:** Antonio Muñoz-Llerena, Pablo Caballero-Blanco, Elena Hernández-Hernández

**Affiliations:** 1Physical Education and Sport Department, University of Seville, 41013 Seville, Spain; 2Research Group “Social Inclusion, Physical Education and Sport, and European Policies in Research” (HUM-1061), University of Seville, 41013 Seville, Spain; 3Research Group “Movement, Intervention Techniques, Values, Learning, Sport and Security (MOTIVA2)” (SEJ-570), Universidad Pablo de Olavide, 41013 Seville, Spain; 4Sport and Computer Science Department, Universidad Pablo de Olavide, 41013 Seville, Spain

**Keywords:** positive youth development, life skills, teaching personal and social responsibility model, sport education model, hybridization

## Abstract

Decision-making is one of the most important life skills for young athletes to succeed in their daily lives and can be improved through Positive Youth Development (PYD) programs. Although the application of this type of programs has increased within educational and recreational sports, there is a lack of research relating PYD through competitive sports. Therefore, the objective of this study was to determine whether an intervention with competitive female youth players improves their decision-making. A hybrid PYD program was applied (37 trainings and 14 games) using convergent mixed methods. 15 girls (8–10 years) and a male coach participated in the study. Semi-structured interviews and field notes were used to collect qualitative data and the Decision-Making Style in Sport questionnaire was used to collect quantitative data. The results showed a positive perception of the improvement in decision-making in both the players and the coach; however, there was no statistical significance between the pretest and the posttest, and the integration between qualitative and quantitative data was mostly discrepant. Despite the lack of significance in the statistical analysis, the findings suggest that the intervention program appears to have beneficial effects on the development of decision-making skills in the players.

## 1. Introduction

Decision-making is known as the cognitive emotional, motivational and volitional conception of the ability to make decisions, which implies elements such as believing in one’s own possibilities, concern about improving this ability and being able to succeed in risky situations [1]. This skill could be considered one of the most important life skills for youth athletes [2,3,4].

Life skills can be defined as those personal characteristics, abilities and assets that can be fostered within sport and transferred to a non-sportive context [5]. These skills can be developed through a continuous and individualized process consisting of internalizing those personal assets within the sport context for their subsequent transfer to other life contexts [6]. The skills allow people to meet the requirements of and succeed in their daily lives (e.g., school, work, home) [7,8], and one of the ways to develop them is through Positive Youth Development (PYD) programs.

The PYD paradigm can be considered a strengths-based approach, in which youth are seen as having resources to develop instead of problems to solve, and its aim is to improve their abilities and their potential for healthy development and for adaptation to real society [9]. Programs guided by PYD are characterized by having a defined framework that helps youth have optimal developmental experiences in organized activities [10], facilitating life skills acquisition [11]. This framework includes, for example, a physically and psychologically secure atmosphere, positive and supportive interpersonal relationships, authentic and challenging activities in which there are opportunities to develop life skills, prosocial norms, or a sense of belonging [12,13].

PYD programs have traditionally been applied through organized leisure activities, in which youth spend almost half their time during the day [14]. Within organized activities, sport is considered the most popular [15] and has been generally considered the best way to promote PYD [16,17,18], given that it can provide opportunities to be responsible and to learn the skills necessary to achieve success in life [8,19,20]. The benefits of participating in a sports PYD program are varied and affect all ecological levels (i.e., personal, interpersonal, school, community, and society) [21]. Character, competence, confidence, connection, health, life skills, and atmosphere can be improved through PYD programs within sport [22].

On a more specific level, life skills development can make a positive impact on the four developmental dimensions: physical, cognitive, psychological/emotional, and social/moral [23]. Within the physical dimension, sports PYD programs can promote physical fitness (e.g., cardiovascular fitness, strength and resistance, flexibility, bone health) [17,24,25,26,27,28] and sport talent [29]. The effects on the cognitive dimension can result in an improvement in cognitive skills, such as self-regulation and decision-making [17,25,27,28,30], and academic and professional performance [5,31,32]. Improvements in the psychological/emotional dimension can be related to better self-esteem, mental health and emotional management [5,17,25,27,28,29,31] and higher self-confidence, self-efficacy, self-determination, motivation, enjoyment, commitment, and empowerment [31]. There may also be positive effects on the social/moral dimension, with a development of empathy, prosocial behavior, leadership, responsibility, communication, conflict resolution, citizenship, social inclusion, fair play, equity, teamwork, or assertiveness [5,17,18,23,25,27,28,30,31,32,33,34].

In order to achieve those developments, in recent years there has been an increasing tendency to utilize pedagogical models to design sports PYD programs. Pedagogical models can be defined as the way to organize interdependent elements involved in the teaching-learning process to achieve specific learning goals [35], being considered as long-term approaches that provide a comprehensive and consistent teaching plan [36]. Two of the most used pedagogical models within sports PYD are the Teaching Personal and Social Responsibility model (TPSR) [19] and the Sport Education (SE) model [37,38].

On the one hand, TPSR, created by Don Hellison, is a values-based way of education and training, which aims to facilitate the learning of values or life skills while trying to achieve sport-specific goals [19]. This model is usually used by sports professionals who try to focus on the integration of socioemotional learning and sports activities, reflecting on socially and personally responsible behaviors both within and outside sport [39,40]. Its main characteristics are strong coach-athlete relationships, empowerment, and gradual personal and group reflections [19].

On the other hand, SE, designed by Daryl Siedentop, aims to provide youth with authentic and enriching sports experiences [37]. This model is underpinned by constructivist learning theories and prioritizes implicit and informal teaching strategies to achieve an autonomous and responsible learning [41,42]. It also has six defining characteristics (i.e., seasons, affiliation, formal competition, keeping records, culminating event, and festivity) arisen from the particularities of sports, as well as long- and short-term goals and specific methodological strategies that allow practitioners to educate participants to become competent, literate and enthusiastic athletes [38].

Despite the benefits these two pedagogical models can bring to athletes, there is not a single model that can be effective in all contexts [43]. This belief, together with the need to overcome their limitations, has led researchers to combine different models, partially or completely [31]. This combination is defined as hybridization [44], and it can be effective in maximizing the impact of models-based programs [45]. Previous research showed different forms of hybridization [46], one of the most widely used being the combination of TPSR and SE models [31].

Pedagogical models and PYD programs have been implemented within three different sports contexts: educational (i.e., physical education), recreational (i.e., out-of-school-time activities) and competitive. Although PYD within educational and recreational sport has been widely studied [17], competitive sport has barely been addressed in PYD research [47], due to the traditional belief of coaches in the incompatibility of life skills development and sports performance [48] and to the differences between the PYD paradigm and the institutional and organizational identity of most sports leagues and clubs at present [49]. However, competitive sport can help youth develop their social capital, contributing to the interiorization of important assets for adult life, such as the importance of success, resilience, performing under pressure, and pulling themselves together after a defeat [50].

Due to the lack of research on PYD through competitive sports and the potentialities and benefits this type of program could provide to athletes, the authors decided to carry out a study in which a hybridized TPSR+SE PYD program was designed and applied in competitive team sports. The main goal of the study was to determine whether an intervention in female youth players belonging to a competitive volleyball team would have a positive impact on the decision-making skills of the athletes. To achieve this goal, a set of research questions and specific goals were established:**Research question 1**: What is the perception of the participants about the development of the decision-making skills of the players?**Specific goal 1**: To evaluate the changes produced in the decision-making skills of the players after the application of the designed PYD program.**Specific goal 2**: To analyze the perception of the players and the coach about the learnings of the former during a PYD intervention.**Specific goal 3**: To determine how the perceptions of the participants are related to the statistical results on decision-making skills.

This study is part of a larger research project, whose aim was to design a hybridized TPSR+SE program [51] and apply it to competitive youth volleyball players in order to assess its effects on peer leadership, personal and social responsibility [52], decision-making and atmosphere, as well as to analyze its implementation fidelity and the perception of the participants (i.e., players and coach) about the structure of the intervention.

## 2. Materials and Methods

### 2.1. Context

The main goal of the intervention was to help the athletes become competent, literate, enthusiastic, and responsible players, capable of leading teams and making decisions for the team’s sake. The program was applied in a subsidized school in Seville (Spain), which belonged to an upper-middle socioeconomic level neighborhood.

In Seville, non-federated sports clubs do not have clear guidelines for coaches on how to coach, especially in grassroots sports. Usually, coaches tend to focus on winning rather than on the personal development of the athletes; however, Catholic Schools Competition teams (where the intervention took place) are generally more flexible and allow coaches to decide the coaching model to follow. This implies that, in order to focus on the holistic development of the athletes, previous training and knowledge of coaches on methodologies that aim at personal and social development become crucial.

In this study, the intervention has been carried out in volleyball, due to its differential features in relation to other team sports since the structure of the game and the rules make volleyball a sport with high temporal demands. The impossibility of maintaining the ball, being allowed to play only with strikes, contributes to an enhancement of technique and ball control, as well as increases the dependency on the team. In volleyball, only three contacts with the ball are allowed, which means that all game actions (except for the serve) are sequential, depending on the previous and the next ones, making teamwork and cooperation essential and forcing players to make decisions in a short period of time. This puts the players into constant stressful situations that they must solve, making these skills important for players. Furthermore, in the initial stages of development within volleyball there is a low technical quality; consequently, tactical cooperative behaviors need to be developed to compensate the limited technical efficacy.

### 2.2. Research Design

In this study, a convergent mixed methods design has been used, combining a descriptive phenomenological design (qualitative) with a pretest-posttest design (quantitative) (Figure 1). This type of mixed methods acknowledges that qualitative and quantitative methods alone are not adequate to capture trends and details in the context of the present research [53,54]. Thus, mixed methods allow us to join both methodologies together in order to make a thorough analysis of the research questions and to enable the strengths of one method to mitigate the weaknesses of the other [53,55].

In the qualitative part, a design based on descriptive phenomenology has been used [56,57]. This design allows researchers to describe the lived experiences of individuals about a phenomenon [53,56,57], with the aim of reaching the essence of their experiences [58]. In the quantitative part, a pretest-posttest design has been utilized to analyze the impact of the independent variable on the participants [59]. The whole study has been carried out under a pragmatic worldview, characterized by focusing on research questions and consequences, using multiple methods to solve the research problem [55]; this worldview is the most appropriate for mixed methods designs [55].

### 2.3. Sample

Intentional, non-probabilistic sampling was used based on the participants and sport clubs that showed interest in and commitment to participate in the study. 15 girls (*n* = 15, age = 8.93 ± 0.80 years) and one coach (male, 26 years old, Physical Activity and Sport Sciences graduate and with four years of experience in coaching youth volleyball) who were enrolled in extracurricular volleyball training and competition in a subsidized school in Seville (Spain) during the 2018/2019 sports season participated in the study. The coach was also the main researcher of this study, adopting a full participation role [53,58]. The team competed in the U-10 category within the Catholic Schools Competition.

Inclusion criteria for the players were: (a) participation in the extracurricular volleyball activity; (b) willingness to participate in the study; (c) written consent of parents/guardians; (d) attendance for at least 80% of the sessions; and (e) not participating in other organized sports activities. Inclusion criteria for the coach were: (a) having carried out pretest and posttest assessments; (b) commitment to implement the intervention program throughout the entire sports season; (c) actively attending initial and continuous training on the program and the two hybridized pedagogical models (i.e., TPSR, SE); and (d) maintaining an adequate implementation fidelity.

This research has followed the ethical guidelines defined in the Declaration of Helsinki regarding consent, confidentiality, and anonymity of the participants through an agreement signed by the school board and the parents/guardians of the players. It has also been approved by the Ethics Committee for Human Research of the Universidad Pablo de Olavide and by the Andalusian Portal of Biomedical Research Ethics (protocol code 2345-N-20).

### 2.4. Variables

The dependent variable was the decision-making of the players. This variable is divided into three different dimensions (perceived decisional competence, commitment to decisional learning, and anxiety and stress when deciding) [1]; only the two first dimensions are analyzed in this study. Perceived decisional competence (PDC) is defined as the perception of the athlete about his/her own ability to make decisions within sport, while commitment to decisional learning (CDL) is understood as the commitment shown by the athlete to improve his/her decisional competence and follow the instructions given by the coach [1].

The independent variable was the TESPODEP program, which is a hybridization of TPSR and SE pedagogical models that focuses on helping athletes become better players by developing life skills such as leadership, personal and social responsibility or decision-making, and transferring them to their daily lives [51]. This program used the general structure of TPSR, complemented by sport-specific elements from SE and employing a combination of methodological strategies that emerged from both models. A description of the TESPODEP program can be found elsewhere [51,52]; however, a brief description of the main characteristics is included as Appendix A.

A total of 37 training sessions and 14 competition games were performed throughout the season, implementing sports content together with responsibility levels. All levels were applied in all sessions and games, but one of them was especially highlighted in each session; in the games, level 5 (transference) was the highlighted one (Table 1).

### 2.5. Data Collection

#### 2.5.1. Semi-Structured Interviews

Semi-structured interviews were used to assess the perception of the players about the development of their decision-making skills. The interviews were designed ad hoc based on other interviews utilized in previous research based on the TPSR model [58,60,61]. There were four different blocks of questions: (1) questions related to previous experience in volleyball and other sports practice (e.g., “Had you practiced volleyball before starting this year?”; (2) questions about perception of learning and skill development (e.g., “Do you think that the training sessions have helped you to make better decisions when you have a goal or when there is a problem? Why?”); (3) questions addressing the perception about the intervention program (e.g., “What do you think about trying to give you autonomy and letting you try yourselves?”); and (4) questions about the coach (e.g., “What do you think about the performance of the coach during the program?”). The full script of the interview is included as Appendix A.

#### 2.5.2. Field Notes

Field notes were taken by the researcher, who was also the coach of the team (adopting a full participation role) [53,58], to assess his perception of the development of the decision-making skills of the players. The structure of the field notes was designed ad hoc based on a diary structure used in previous studies [62], including four different sections: (1) general information about the session; (2) responsibility level and goals; (3) methodological strategies and foundations utilized; and (4) observations.

#### 2.5.3. Decision-Making Style Questionnaire

To assess decision-making, the Decision-Making Style in Sport questionnaire (in Spanish, Cuestionario de Estilo de Toma de Decisión en el deporte, CETD) [1] was used. This questionnaire was validated in the Spanish context by the previous authors and consisted of 30 items divided into three factors, each of which was made up of 10 items: PDC (α = 0.82), anxiety and stress when deciding (α = 0.83) and CDL (α = 0.75). Due to the objectives of this study, only the factors of PDC and CDL were evaluated. All items were answered through a 4-point Likert-type scale ranging from 1 (totally disagree) to 4 (totally agree).

#### 2.5.4. Procedure

For sample selection, a meeting was held with the school board to explain the project and its benefits to girls. After the approval of the board, an informative meeting was conducted with the parents of all the players, in which the characteristics and procedure of the intervention program were explained to them; the same protocol was followed with the players, emphasizing the voluntary nature of participation.

After receiving the informed consent of parents and players, coach training was carried out. The coach, who was the main responsible for designing the TESPODEP program, learned the fundamentals of the TPSR and SE models before the design of the hybridization. After the design of the intervention program, and following different authors [34,63], one month prior to the beginning of the implementation, the coach received an initial training of 30 h within one week to refresh the theoretical foundations, goals and relevant methodological aspects of the TPSR and SE models given by two experts from each (i.e., co-authors). After initial training, the program design was revised with minor modifications. During the program, monthly meetings (except for the first month, when they were weekly) were held to maintain continuous training and to progressively analyze the fidelity of the implementation [64], which was considered adequate after its analysis.

The interviews were carried out throughout the two weeks after the end of the intervention, after the training sessions and without setting a time limit, collecting between three and four interviews daily; the protocol proposed by [53] was followed. The recordings were made with an iPad Air 2 device and transcribed verbatim. Their length was between 50 and 75 min. Field notes were taken between 30 and 90 min after each training session, written on a record sheet using Microsoft Word software.

The CETD questionnaire was applied the week before the start of the intervention (pretest) and the week after the end of the intervention. The questionnaires were delivered to the players and completed at the beginning of the training session, under the supervision of the coach and first author of this study. Before the completion, the instructions were explained, emphasizing the maintenance of a relaxed atmosphere to improve the concentration and precision of the answers [34,54,65]; confidentiality and anonymity were ensured, asking for honesty in the answers, and all questions that arise during the process were solved.

### 2.6. Data Analysis

#### 2.6.1. Qualitative Data Analysis

Qualitative Content Analysis (QCA) [66] was the methodology used to analyze both interviews and field notes (Figure 2). Within QCA, theme analysis technique was applied, carrying out two different analytical processes: deductive category assignment (Figure 3A), in which categories are created in advance based on theoretical foundations and the interviews/field notes; and, subsequently, inductive category formation (Figure 3B), in which the already existing categories are synthesized.

Different techniques were utilized to ensure methodological rigor (i.e., data reliability, transferability, dependability, and confirmability) [67,68,69]:To achieve reliability, the methodological integrity procedure [70] was followed. Within this procedure, data and analyst triangulation [53,58] and critical friend [71] techniques were applied (Table 2).To ensure transferability, rich and thick descriptions were used to convey the findings [53].To guarantee dependability, data and analyst triangulation and rich, thick descriptions were utilized.To attain confirmability, analyst triangulation, critical friend, audio recording and verbatim transcription, and textual quotes from players and coach were used.

Analyst triangulation was carried out by the first and second authors, researchers with experience in the field of study. The procedure established by [72] was followed to objectively calculate intercoder reliability (i.e., Cohen’s kappa and the percentage of agreement). The analysis showed an intercoder reliability of 0.950 and 0.952 (kappa) and 99.85% and 100% (agreement) for the interviews and field notes, respectively.

Qualitative data analysis was carried out using NVivo^TM^ 12 Plus.

#### 2.6.2. Quantitative Data Analysis

The first step was to introduce the data collected from the CETD questionnaire into a SPSS database. After creating the database, outliers and missing values were looked for; there were no outliers or missing values. Subsequently, reliability analysis tests were applied: Cronbach’s α and McDonald’s ω [73,74] (Table 3); adequate values for reliability were found, at ≥0.70 for both α [75] and ω [76]. Afterwards, descriptive statistics were obtained and normality tests were carried out (i.e., Kolmogorov–Smirnov and Shapiro–Wilk tests; graphical distribution with histograms, P-P and Q-Q plots) [77]. Finally, since the sample did not follow a normal distribution, non-parametric tests (i.e., Wilcoxon test for related samples) were applied.

For the normality analysis (skewness, kurtosis and Kolmogorov–Smirnov and Shapiro–Wilk tests), reliability (Cronbach’s α), descriptive statistics and non-parametric tests, SPSS 26.0 was utilized. To calculate McDonald’s ω, “psych” 2.1.9 package [78] was used in R 4.1.2.

#### 2.6.3. Data Integration

Data were integrated according to the procedure proposed by [55]. This procedure was represented through a joint display [79] and consisted of six different steps:Obtaining results from the analysis of qualitative and quantitative data.Looking for common concepts in the results.Comparing both types of results for each common concept.Determining how the results are congruent, discrepant or complement each other.Interpreting and resolving differences.Utilizing different procedures for data transformation.

The methodological rigor criteria for data integration were based on [55]. Qualitative and quantitative data collection and analysis have been carried out rigorously, in response to hypotheses and research questions; both types of data have been integrated effectively; these procedures have been organized into a specific research design that allowed the study to be put into practice in a logical way; and the whole process has been framed within a specific theoretical and philosophical framework.

## 3. Results

### 3.1. Interviews

As previously explained, a first deductive coding was carried out to determine the main categories of decision-making, creating the categories of PDC and CDL. Subsequently, an inductive coding of the two main categories was performed, establishing the subcategories shown in Table 4.

#### 3.1.1. Perceived Decisional Competence

This category refers to the perception of the players about their PDC. The subcategories formed in the theme analysis were general decision-making, achieving goals, sport decisions and problem solving.


**General decision-making**


This subcategory alludes to the PDC related to decision-making at a general level, outside of the rest of the subcategories. Only four players referred to general decision-making, all of them considering that they enhanced their decision-making because of participating in the program. This implies a higher number of decisions made and its transference outside of the team.

P03: “Now I make more decisions than before, because previously I was not making any decision, but now I say ‘I am going to try to do this because that’ [sic]. … I could also do what I felt would be best [when acting as a coach]. Also, in my life out of volleyball, I said ‘Look, this could be better for everyone, so I’m doing it’.”

For these players, the intervention program was useful in acquiring the skills and competencies necessary to be able to make decisions on a general level, and now they make more decisions and in more contexts of their lives.


**Achieving Goals**


This subcategory could be defined as the PDC related to goal setting and achievement of these goals. 11 players considered that they had learned to make decisions to achieve their own goals, both within and outside sport.

P14: “What I do is to propose myself something, and I do it. I mean, I proposed myself to tidy up my bedroom, and I did it, I tidy it up every day.”

One of them also explained how she divided her goals into smaller ones to achieve them progressively, while another showed how she had reached the goals she had set during the intervention.

P02: “I have [enhanced decision-making] because I get less angry and my serve passes the net, and because [I have achieved] some goals, for example, studying math.”

P03: “Look. If I try to make a serve, I say I must put my hand hard, … not soft. And focus. I focus a lot.”

On the contrary, four athletes believed that they had not improved their decision-making to achieve goals because they did not achieve the goals that they set themselves during the intervention, when the focus was on the level 3 of responsibility.

P15: “So so [sic]. … Because, of the goals that I set before, I have achieved none.”

Most of the team thought that they had learned to make decisions that help them achieve goals, although four of the players considered the opposite.


**Sport decisions**


This subcategory refers to PDC within the sport context, during training sessions or games. Half of the players referred to this subcategory: four of them considered that they had improved, while the other three believed they had not become better at it, although one of them spoke about having learned to make decisions when setting a technical goal (i.e., serving better).

P03: “Look. If I try to make a serve, I say I must put my hand hard, … not soft. And focus. I focus a lot.”

P14: “I have not [improved]. … [Because] Sometimes I move out [when the ball comes], but not always. … Because I say ‘Wow, the ball, what should I do?”

In general, approximately a third of the team believed that they improved their decision-making within the sport context, while one fifth considered the opposite.


**Problem solving**


This subcategory relates to PDC when solving problems that arise in training sessions or in daily life. Nine athletes considered that they enhanced their decision-making with respect to solving conflicts near them, either when they are involved, when they mediate a conflict between others or when a problem affects the entire team.

P06: “[Thanks to the program] I can make decisions to solve conflicts between others. Because I have solved conflicts, and now if someone gets angry, I can solve them.”

On the other hand, four players thought that they had not got better because they did not transfer it to their daily lives or because they only made the problem worse when trying to solve it.

P09: “I have not [enhanced my decision-making for solving problems]. … Well, it depends. … Because the ideas I have always end up making it worse. Because unintentionally I say bad things that the other party said to them.”

Similarly to the subcategory of achieving goals, most of the athletes believed that they had improved in making decisions to solve conflicts and problems; only four players disagreed with that statement.

#### 3.1.2. Commitment to Decisional Learning

This category refers to the perception of the players about their own CDL. There were no subcategories formed within this category, and athletes alluded to this category in relation to autonomous goal setting as a form of commitment in learning how to make decisions.

P14: “I have proposed myself not arguing that much with my sister. … I have told myself that, to achieve a goal, I must make something bigger than the goal I have set. For example, if I want to make a serve that passes the net, … what I do is to try to pass five, not one. … I do not set myself a lot [of goals], but at least I have set the goal of tidying up my room every day.”

Some of the players demonstrated their CDL development by setting their own goals and deciding their own course of action. However, none of them referred to following the instructions given by the coach.

### 3.2. Field Notes

The same procedure as in the interviews was followed in the analysis of the field notes (deductive coding followed by inductive coding). However, only one main category was formed, since the coach did not refer frequently to the decision-making of the players (Table 5).

#### Perceived Decisional Competence

This category refers to the perception of the coach about the decisional competence of the players. The coach believed that the players demonstrated an improvement in decisional competence when they performed responsibility roles.

2nd session: “[The coach] Has been able to solve a conflict between three players autonomously, showing a development in her leadership, decision-making and conflict resolution skills.”

At a collective level, the team was able to enhance their decision-making by looking for the most beneficial option for the team. Moreover, the coach favored the autonomous work of the players, letting them make the decision to set and achieve their own goals.

19th session: “In the group reflection … they are told to bring a new goal decided by themselves.”

21st session: “They are given the option of a real game for the last 25′ if they can overcome a challenge. They choose one player to try, and she succeeds.”

### 3.3. Decision-Making Style Questionnaire

The descriptive statistics of the CETD questionnaire are shown in Table 6. The Wilcoxon test for related samples (Table 7) did not show any significant differences in either of the factors between the pretest and the posttest. It could be stated that the intervention program did not cause significant changes in the decision-making of the players.

### 3.4. Integration

The main categories arising from QCA in both interviews and field notes (i.e., PDC, CDL) coincided with the two factors of the CETD questionnaire. Therefore, a comparison between homonymous categories and factors was carried out through a joint display (Table 8).

The results of the integration showed that most of the qualitative data were discrepant with the quantitative data. The only congruent comparison consisted of a minority of the results of the PDC category within the interviews, where a reduced number of players considered that there was a lack of development in their decisional competence. Nevertheless, the majority of players thought they had improved their decisional competence and CDL; similarly, the coach also believed in the enhancement of the decisional competence of the team, although he did not state any information on CDL in the field notes.

## 4. Discussion

The main objective of this study was to determine whether an intervention in female youth players belonging to a competitive volleyball team would have a positive impact on the decision-making skills of the athletes. To give an answer to this goal, it is necessary to reflect on the impact of the program on the athletes (including the perceptions of the players and the coach about decision-making development of the former and the statistical results of the questionnaires) and think about how perceptions and statistics relate to each other.

### 4.1. Perceptions of Players and Coach about Decision-Making Development

The analysis and interpretation of the perceptions of the participants (youth and group leader) about the learnings acquired in the intervention have gained attention in the scientific literature in the field of PYD through sport with respect to TPSR, SE or hybrid interventions [32,80,81,82]. PYD allows participants to understand their experiences and figure out why and how the intervention program worked and what consequences it has for their lives [58].

Regarding the first research question of the study (what is the perception of the participants about the development of the decision-making skills of the players?) and the second specific goal (to analyze the perception of the players and the coach about the learnings of the former during a PYD intervention), the results showed that the participants perceived a positive improvement in the decision-making of the players throughout the intervention. This enhancement was related to their decisional competence, considering that they were able to make a higher number of decisions both within and outside sport, as well as they learned to make decisions that benefited the whole team. Making decisions to solve conflicts was the improvement cited most often, followed by better decision-making for achieving goals or sports-specific decision-making. However, a small part of the team believed that they did not improve in one (or a few) of the aforementioned aspects. Regarding CDL, many players seem to have developed it by setting their own goals and choosing their own path to achieve them.

The perceptions of the athletes are consistent with previous literature that applied TPSR, SE or their hybridizations in volleyball [80,83] or other sports [41,84,85]. Similarly, the perceptions of the coach are in line with other research based on these pedagogical models in the educational context [85,86,87], although, to the knowledge of the authors, there is no existing literature that explicitly analyzes the perception of the coach about the development of decision-making within competitive sport.

The perceived evolution of the decision-making skills of the athletes could be considered as a life skill acquisition that allows them to successfully meet the demands of their daily lives [8], and could be attributed to different features of the intervention program. On the one hand, including responsibility levels 3 (autonomy) and 4 (leadership and helping others) helped to create an optimal context for decision-making development, putting the players in charge of their peers during training sessions; this is closely related to two of the methodological foundations (i.e., integration and empowerment).

On the other hand, the presence of the methodological strategies of fostering social interaction, setting roles and tasks, leadership, and giving voice may also have contributed to the enhancement of decision-making skills of the players. Organizing the tasks to facilitate social interaction and to give voice to the players, as well as setting responsibility and leadership roles, allowed the athletes to have more freedom throughout the intervention, giving importance to their opinions and suggestions and normalizing cooperation, teamwork and conflict resolution, including all the decisions that they must make.

Another relevant feature that may have positively impacted decision-making is the use of the specific methodological strategies of independent practice, conflict resolution and team identity. Independent practice, together with the strategies for setting roles and tasks and leadership, implied that the players who performed the roles of coach and assistant coach led the team throughout the week, making the corresponding decisions under the supervision of the adult coach. Conflict resolution also allowed athletes to make decisions to solve problems that arise in training sessions or games, looking for what is best for the team and putting personal preferences aside. Additionally, creating a team identity required a collective decision-making about the name of the team, the battle cry and the team songs to sing from the bench during the games.

The last feature that may have contributed to fostering decision-making is volleyball itself. The high temporal and technical demands derived from the impossibility of retaining the ball cause a decrease in the reaction time, which is smaller than in other sports. This favors the development of cooperation, autonomy and decision-making in the players.

### 4.2. Effects of the Program on Players

The effects of TPSR, SE or hybridized PYD programs on participants have been widely studied in the educational context [61,65,88]; however, there are only a few studies addressing competitive sport [89,90].

Regarding the first specific goal (to evaluate changes produced in the decision-making skills of the players after the application of the designed PYD program), the results showed a slight increase in the median values of both factors (PCD, CDL) at the end of the intervention, although it was not significant. Therefore, there were no significant changes in decision-making after the intervention.

To the knowledge of the authors, there are no studies that used the CETD questionnaire to assess decision-making through TPSR, SE or hybridized pedagogical models, as most studies that utilize these models and analyze decision-making focus on sports decisions using observational methods [41,80,83] or on the psychosocial aspects of decision-making using qualitative methods [85,86,87]. Due to the lack of research addressing decision-making from a psychosocial perspective using quantitative methods, it is not possible to compare these results with the existing literature.

It is important to reflect on why there is a lack of significant changes in the statistical analyses. Following [91,92], there are four cognitive processes when answering a question: (1) understanding the question; (2) looking for information in the memory to answer; (3) reflecting on the information needed to answer; and (4) communicating the answer. However, a strategy called satisficing takes place when a different process is followed to give an answer [93]. According to the last author, this can happen for different reasons: a poor design of the items, a limitation of cognitive capacities of the participant, boredom, lack of interest, or a harsh context/environment. Taking into account all the possible reasons, the authors believe that the context might have affected the results of the posttest: the pretest was applied in an outdoor court at 16:00 h in November, when the environmental conditions are comfortable in Seville; however, the posttest was carried out at the end of May in the same conditions except for the temperature (40 º C), which could have affected the quality of the data. The sample size might also have played an important role in the lack of statistical differences: 15 individuals could not be enough for the statistical procedures to reflect changes, since *p*-values are directly related to sample size (i.e., a larger sample size enables a smaller standard error and confidence interval) [77].

### 4.3. Data Integration: Relationship between Perceptions and Statistics

Integration in mixed methods research is where qualitative and quantitative data interact with each other, enabling a deeper understanding of both types of data separately [55]. In this study, integration was carried out by comparing both results and checking their agreement, using a joint display [79].

Regarding the third specific goal (to determine how the perceptions of the participants are related to the statistical results regarding decision-making skills), most of the qualitative and quantitative results were discrepant with each other, since qualitative data showed an improvement in decision-making while quantitative data showed no significant changes. The only congruent comparison observed was in the PDC, where a few players referred to a lack of development in their decisional competence. However, it is recommended to be cautious before drawing conclusions due to the relativity of the results [94]: the majority of the group seemed to have enhanced their decision-making, but not all the players improved all the abilities and skills intrinsic to decision-making or are able to use all of them in all situations.

Ensuring that the integrated data are 100% congruent is a hard task [95,96]. In the present study, qualitative coding was carried out on the basis of a general conception of the dimensions and factors (i.e., PDC, CDL) for deductive coding. After that, the inductive coding delved into the dimensions, helping the qualitative analysis encompass deeper and more specific features within PDC and CDL, allowing the authors to discern learnings of the players that may not be expressed through a questionnaire. All of the above could make it difficult to reach a majority of congruent results. Therefore, and following the recommendations of [55] on convergent mixed methods, the authors are more confident in the qualitative results for two different reasons: (1) the possible satisficing in the answers of the questionnaire; and (2) the small sample.

### 4.4. Theoretical and Practical Implications

Although it is not a panacea for a society that has a greater and greater lack of values and life skills, using this type of program in competitive sports clubs and teams may have a positive impact in the close context of the participants, which, according to chaos theory, could imply a greater social change in the future. In this way, this study contributes to filling the existing gap in the scientific literature on PYD through competitive sports, as well as in mixed methods research in the field of competitive sports interventions, by applying a hybrid program that could serve as a reference for future PYD-based interventions in team sports.

From a theoretical perspective, this work provides understanding on competition seen through the PYD paradigm, encompassing how competitive sports participation may help to achieve life skills acquisition (and, specifically, decision-making) in athletes. The fact that the main goal of competitive sports is talent development and performance implies that training programs are focused on the empowerment and the development of personal and social skills of the athletes, features that are inherent to PYD. Therefore, this study sheds light on how life skills development and sports (i.e., technical, tactical and physical) development are compatible.

From a practical perspective, this work provides confirmed strategies for practitioners related to task design and application, within a methodological structure for developing and transferring life skills through competitive sports. Some recommendations for practitioners to design training tasks and programs are:Adapting the methodological structure to the athletes and the context of intervention.Setting goals related to sports and life skills.Integrating PYD-related strategies into games and training tasks.Using the different methodological strategies to facilitate PYD development and foster life skills acquisition.Involving all athletes in all responsibility roles throughout the season, allowing them to make their own decisions.Maintaining a balance between performance and PYD intervention.

### 4.5. Strengths and Limitations

This study presents several strengths: the high commitment of the coach, who was also the first author of this study; the presence of initial and continuous coach training, which helped him consolidate his knowledge about TPSR and SE models; the length and consistency of the intervention, which consisted of 37 training sessions and 14 games throughout eight months, while most PYD interventions using TPSR or SE have a duration between eight and 20 sessions [42,44]; the use of mixed methods, which contributed to a holistic and thorough understanding of the intervention and its results; and the methodological framework of TESPODEP, which facilitated life skills (and, especially, decision-making) development, both within and outside sports.

There are also some limitations to this study. On the one hand, the sample size is appropriate to obtain relevant information at a qualitative level, but it seems to be insufficient to achieve significant statistical results [53,55]. On the other hand, there was only one experimental group enrolled in the study, of a specific sociocultural and socioeconomical background, sport, competitive level, age, and sex. Another limitation was the lack of a control group, which could have helped reveal more relevant quantitative results. Furthermore, the coach did not have previous experience with the application of PYD programs, although he knew the theoretical foundations of both the TPSR and SE models. Lastly, only the coach and the players participated in the study, without including key social agents such as parents, teachers, or peers.

In order to alleviate these limitations, there could be future lines of research based on this work:Including a larger sample and different experimental and control groups, with different ages, sexes and competitive, sociocultural and socioeconomic levels, and within different sports.Applying the intervention throughout more than one sports season, assessing long-term improvements.Increasing the number of data sources, including key social agents (e.g., parents, peers, teachers).

## 5. Conclusions

The purpose of the study was to determine whether an intervention in female youth players belonging to a competitive volleyball team would have a positive impact on the decision-making skills of the athletes. Based on the results obtained and the reflections made, the TESPODEP program seems to have beneficial effects on developing decision-making skills of the players and, if carried out properly, the program can foster life skills acquisition in the participants.

## Figures and Tables

**Figure 1 ijerph-19-13261-f001:**
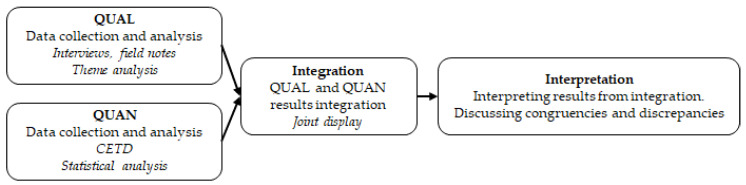
Procedure followed in the convergent mixed methods design.

**Figure 2 ijerph-19-13261-f002:**
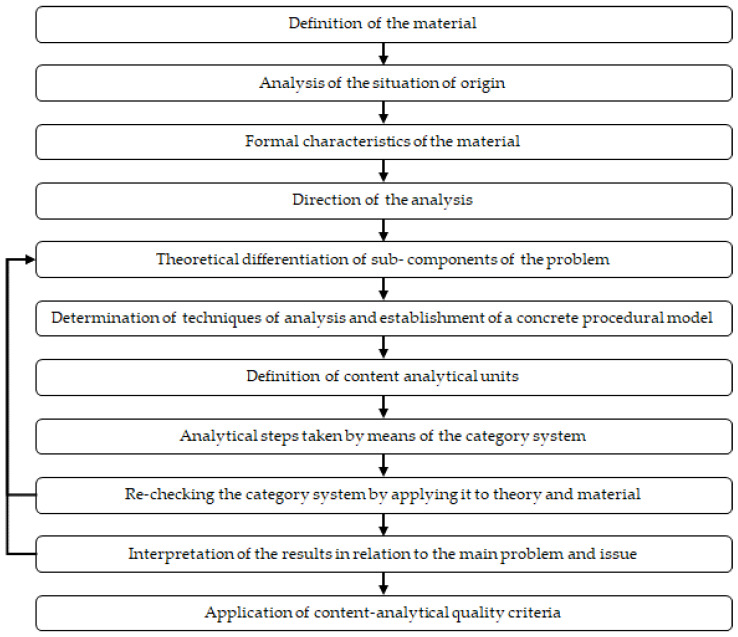
Steps of QCA.

**Figure 3 ijerph-19-13261-f003:**
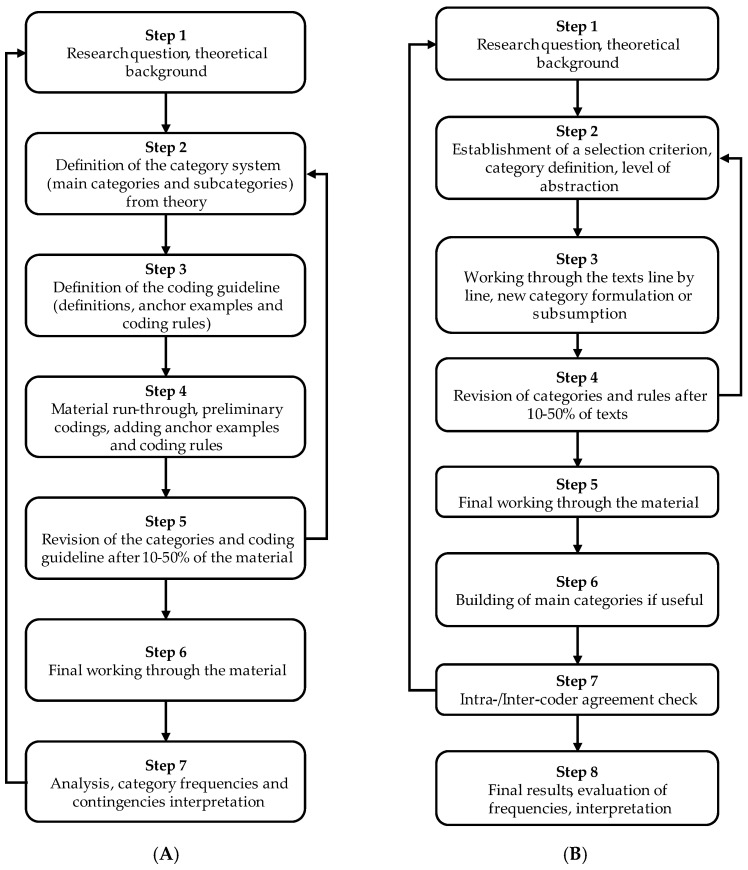
Theme analysis procedure. (**A**) deductive category assignment; (**B**) inductive category formation.

**Table 1 ijerph-19-13261-t001:** Sessions, contents and levels throughout the program.

S	Content	RL
1	Overhead pass-T; Underarm pass-T	I
C	Competition	5
2	Overhead pass-T; Underhand serve-Sec	1
3	Underarm pass-T; Underhand serve-Sec	1
C	Competition	5
4	Overhead pass-T+M; Underarm pass-T+M	1
5	Overhead pass-T+M; Underarm pass-T+M	1
6	Overhead pass-P+M; Underhand serve-Sec	2
7	Overhead pass-T+M; Underarm pass-T+M; Underhand serve-Sec	2
C	Competition	5
8	Overhead pass-P+M; Underarm pass-T+M; Underhand serve-Sec	2
C	Competition	5
9	Overhead pass-T; Underarm pass-T	1, 2
10	Overhead pass-T; Underarm pass-T; Underhand serve-Sec	2
11	Overhead pass-T+M; Underarm pass-T+M	2
12	Overhead pass-T+M; Underarm pass-T+M; Underhand serve-Sec	2
C	Competition	5
C	Competition	5
13	Overhead pass-T+M; Underarm pass-T+M; Underhand serve-Sec	3
14	Overhead pass-T+M; Underarm pass-T+M; Underhand serve-Sec	3
C	Competition	5
15	Overhead pass-T+M; Underarm pass-T+M; Underhand serve-Sec	3
C	Competition	5
16	Overhead pass-P; Underarm pass-P; Underhand serve-Sec	3
17	Overhead pass-P; Underarm pass-P	3
18	Underarm pass-P; Underhand serve-Sec	3
19	Overhead pass-P; Underarm pass-P; Underhand serve-Sec	3
C	Competition	5
20	Overhead pass-P; Underarm pass-P; Underhand serve-Sec	3
21	Underarm pass-P; Underhand serve-Sec	3
C	Competition	5
22	Overhead pass-P; Underarm pass-P	3
23	Overhead pass-Setting; Underarm pass-P	3
24	Overhead pass-Setting; Underarm pass-P; Underhand serve-P	3
C	Competition	5
25	Overhead pass-Setting; Underarm pass-Receiving; Underhand serve-P	4
26	Overhead pass-Setting; Underarm pass-Receiving; Underhand serve-P	4
27	Overhead pass-Setting; Underarm pass-P	4
28	Overhead pass-Setting; Underarm pass-Receiving; Underhand serve-P	4
C	Competition	5
29	Overhead pass-Setting; Underarm pass-P	4
30	Underarm pass-Receiving; Underhand serve-P	4
31	Overhead pass-Setting; Underarm pass-P	4
32	Overhead pass-Setting; Underarm pass-Receiving; Underhand serve-P	4
C	Competition	5
33	Overhead pass-Setting+attack (jump); Underhand serve-P	4
34	Overhead pass-P; Underarm pass-P; Underhand serve-Sec	4
35	Overhead pass-P+attack (jump)	4
C	Competition	5
36	Underarm pass-Receiving+defense; Underhand serve-P	4
37	Overhead pass-P+attack (jump)	4

Note: RL, Responsibility Level; I, Introduction; S, Session; T, Technique; M, Movement; Sec, Security; P, Precision; C, Competition.

**Table 2 ijerph-19-13261-t002:** Methodological integrity procedure.

Fidelity		Utility	
**Adequate data**	Triangulation was carried out, collecting data from two main sources (players and coach)	**Contextualization of data**	Information regarding the characteristics of the participants and researchers who have been part of the study has been presented
**Perspective management in data collection**	The originating paradigm has been explained, in addition to the second author acting as a “critical friend” to avoid distorting the data	**Catalyst for insight**	Interviews and field notes have been used as a form of data collection, after reflection by the authors on the best way to generate information that allows in-depth analysis
**Perspective management in data analysis**	Analyst triangulation was used to ensure the fidelity of the analyzed results	**Meaningful contributions**	The study has had a positive impact on the development of the participants in the variables analyzed, in addition to showing the scientific community the effectiveness of the implemented program
**Groundedness**	The results obtained were analyzed and discussed, making reference to quotations and reflections	**Coherence among findings**	The data obtained are integrated to show the relationship between the different results, including reflection on the congruencies and discrepancies

**Table 3 ijerph-19-13261-t003:** Reliability of CETD questionnaire.

	*α*—Pretest	*ω*—Pretest	*α*—Posttest	*ω*—Posttest
**PDC**	0.73	0.81	0.57	0.80
**CDL**	0.81	0.91	0.62	0.85

Note. PDC, Perceived Decisional Competence; CDL, Commitment to Decisional Learning.

**Table 4 ijerph-19-13261-t004:** Categories and coding frequency—interviews.

	Perceived Decisional Competence (115)				Commitment to Decisional Learning
	General Decision-Making	Achieving Goals	Sport Decisions	Problem Solving	
Coding frequency	10	59	15	31	18

**Table 5 ijerph-19-13261-t005:** Categories and coding frequency—field notes.

	Perceived Decisional Competence
Coding frequency	3

**Table 6 ijerph-19-13261-t006:** Descriptive statistics of CETD questionnaire.

	N	M	SD	Skewness	Kurtosis
PDC—Pretest	15	2.38	0.54	0.64	−0.88
PDC—Posttest	15	2.42	0.41	0.20	−0.95
CDL—Pretest	15	3.16	0.56	−1.21	2.25
CDL—Posttest	15	3.35	0.36	0.13	−0.65

Note. N, sample; M, Mean; SD, Standard Deviation; PDC, Perceived Decisional Competence; CDL, Commitment to Decisional Learning.

**Table 7 ijerph-19-13261-t007:** Comparation of decision-making factors between pretest and posttest.

	*T*	Median		*z*-Score	*r*	*p*
		Pretest	Posttest			
PDC	54.50	2.20	2.30	0.63	0.12	0.527
CDL	73.00	3.20	3.30	0.74	0.14	0.459

Note. T, test statistic; *r*, effect size; *p*, asymptotic significance; PDC, Perceived Decisional Competence; CDL, Commitment to Decisional Learning.

**Table 8 ijerph-19-13261-t008:** Data integration.

**Factor**		**PDC**		**CDL**	
		Mdn Pre-Post	*p*	Mdn Pre-Post	*p*
		2.20–2.30	0.527	3.20–3.30	0.459
**MC**		**PDC**		**CDL**	
I	*Cg*	P07: “I haven’t [learned to make better decisions] … because I am not good at it”		No data available	
	*Ds*	P03: “Now I make more decisions than before, because previously I was not making any decision, but now I say ‘I am going to try to do this because that’ [sic]. … I could also do what I felt would be best [when acting as a coach]. Also, in my life out of volleyball, I said ‘look, this could be better for everyone, so I’m doing it’”		P14: “I have proposed myself not arguing that much with my sister. … I have told myself that, to achieve a goal, I must make something bigger than the goal I have set. For example, if I want to make a serve that passes the net, … what I do is to try to pass five, not one. … I do not set myself a lot [of goals], but at least I have set the goal of tidying up my room every day”	
FN	*Cg*	No data available		No data available	
	*Ds*	2nd session: “[The coach] Has been able to solve a conflict between three players autonomously, showing a development in her leadership, decision-making and conflict resolution skills.” 21st session: “They are given the option of a real game for the last 25′ if they can overcome a challenge. They choose one player to try, and she succeeds.”		No data available	

Note. PDC, Perceived Decisional Competence; CDL, Commitment to Decisional Learning; Mdn, Median; MC, Main categories; I, Interviews; FN, Field Notes; Cg, Congruent; Ds, Discrepant.

## Data Availability

The data presented in this study are available on request from the corresponding author. The data are not publicly available due to privacy restrictions.

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
