# Peer review of "Fostering Youth Female Athletes’ Decision-Making Skills through Competitive Volleyball: A Mixed Methods Design"

_ijerph, 2022, doi:10.3390/ijerph192013261_

Round 1
Reviewer 1 Report
The authors addressed a sound and interesting subject. The purpose of the paper is clearly stated and the authors used approapriate research methods. The paper is well documente and the authors used references that are relevant for the type of subject addressed. I appreciated that the authors described in the Conclusions section the limitations of their research and that they also provided some future research directions. However, I recommend to the authors to describe in a more clear manner, in the conclusions section the practical and theoretical implications of their study.
Overall, the paper is well -written and I congratulate the authors on their interesting paper.
Author Response
Dear reviewer,
Thank you for your kind words and your suggestion about this research, we are glad to hear that other experienced researchers find this study useful.
Regarding your recommendation, we have modified the theoretical and practical implications section, we hope that now is easier to understand (lines 615-645).
Again, thank you very much for your time and effort reviewing our manuscript. Best regards.
Reviewer 2 Report
This study investigated whether an intervention in competitive female youth players improves decision-making. The author should revise some grammar problems. And some other revisions should be made as follows:
Lines 28 – 30, some keywords are a bit long, please consider using shorter ones.
Lines 42, delete “of”.
Lines 153, the author needs to explain why this study investigated the intervention approach in volleyball activities.
Lines 386 – 387, please double-check the format and aligned it with the other paragraph text, the same thing for the following items.
The conclusion is a bit long, the author may need to consider cutting it more concisely.
Author Response
Dear reviewer,
Thank you for your suggestions and considerations about this research, they have been useful in improving the quality of the manuscript. Now, I will try to answer you in the best way possible your comments.
- The author should revise some grammar problems
- A thorough revision of the whole document has been made by a native-level English professor, and some changes have been made throughout the manuscript.
- Lines 28 – 30, some keywords are a bit long, please consider using shorter ones.
- According to your comments and the comments of other reviewer, keywords have been changed. We have deleted “teaching and coaching innovation” and “innovative pedagogical approaches”, leaving only “positive youth development” and “life skills” (both related to the theoretical framework); and “teaching personal and social responsibility model”, “sport education model” and “hybridization” (related to the methodological framework: they are the two pedagogical model hybridized, and those keywords are frequent in related research).
- Lines 42, delete “of”.
- “Of” has been deleted.
- Lines 153, the author needs to explain why this study investigated the intervention approach in volleyball activities.
- The explanation for the election of volleyball has been modified (Lines 153-166)
- Lines 386 – 387, please double-check the format and aligned it with the other paragraph text, the same thing for the following items.
- Those lines (and the other quotes from the interviews and field notes) were adjusted to the “citation” style of the IJERPH template. We have changed it and added an indent left and right, following journals’ guidelines (Lines 375-465)
- The conclusion is a bit long, the author may need to consider cutting it more concisely.
- The conclusion has been shortened (lines 675-681).
We hope to have answered all your considerations. Again, thank you very much for your time and effort in reviewing our manuscript. Best regards.
Reviewer 3 Report
Frankly speaking, the present work is a highly meaningful and interesting research. The authors attempt to explore whether the youth female athletes’ decision-making skill can be improved by competitive volleyball sports by using a mixed methods design. At last, some imperative and valuable findings can be found by us. Even though, in the section of quantitative statistics, there isn’t a statistically significance between the pretest and posttest, it is lucky that a promising tendency between them has emerged already, and the qualitative results also effectively give some reasonable and persuasive explanations to this deficiency. Perhaps, due to the lack of enough sample size, or the shortage of intervention time, the final result may not be satisfied, but undoubtedly, this is also very crucial and significant for the future study about this theme. Except for those mentioned above, several detailed flaw needed to be taken into account and then make the corresponding revision. Thank you !
1.The title can be revised as follows:“Fostering youth female athletes’ decision-making skills through competitive volleyball sports: A mixed methods design”.
2.The keywords can be corrected as: “positive youth development; decision-making skill; youth volleyball athletes; competitive sports; the mixed method design”.
3.The present study contains a research question, a research hypothesis, and three specific goals, but I think it can be viewed as an exploratory study, so the hypothesis can be deleted and the research question and specific goals should be retained.
4.The conclusion should be refined and concise, instead of redundant and excessive, and exactly shows the main content about the current study. In addition, the content about strengths and limitations should be putted in the section of discussion.
5.Some references are not listed by the corresponding norms about the present journal, so the author should make careful and serious revision for this section.
Author Response
Dear reviewer,
Thank you for your suggestions and considerations about this research, they have been insightful and useful for improving the quality of the manuscript. We are glad to hear that other experienced researchers find this study useful.
Now, I will try to answer in the best way possible your comments.
- The title can be revised as follows: “Fostering youth female athletes’ decision-making skills through competitive volleyball sports: A mixed methods design”.
- The title has been changed (Lines 2-3)
- The keywords can be corrected as: “positive youth development; decision-making skill; youth volleyball athletes; competitive sports; the mixed method design”.
- According to your comments and the comments of other reviewer, keywords have been changed. It is usually advised not to repeat keywords that are already included in the title, that’s why we did not include “decision-making skill”, “youth volleyball athletes”, “competitive sports” and “mixed method design”. Instead, we have deleted “teaching and coaching innovation” and “innovative pedagogical approaches”, leaving only “positive youth development” and “life skills” (both related to the theoretical framework); and “teaching personal and social responsibility model”, “sport education model” and “hybridization” (related to the methodological framework: those are the two pedagogical model hybridized, and those keywords are frequent in related research).
- The present study contains a research question, a research hypothesis, and three specific goals, but I think it can be viewed as an exploratory study, so the hypothesis can be deleted and the research question and specific goals should be retained.
- Hypothesis 1 has been deleted (Lines 123-128), and the discussion about the hypothesis has been modified (lines 560-566).
- The conclusion should be refined and concise, instead of redundant and excessive, and exactly shows the main content about the current study. In addition, the content about strengths and limitations should be putted in the section of discussion.
- The conclusion has been shortened (lines 675-681), and strengths and limitations have been included in the discussion section (lines 646-674).
- Some references are not listed by the corresponding norms about the present journal, so the author should make careful and serious revision for this section.
- Thank you for the thorough revision of the reference list. We used Mendeley Reference Manager to include the reference list, there must have been an error when including the references in Mendeley. We have checked every reference from the list.
- References 1, 10, 14, 21, 28, 30, 33, 35, 39, 47, 50, 62, 64, 65, 70, 72, 76, 82, 83, 84, 88, 89, 90, 93, 94: Journal title was abbreviated
- References 7, 9, 12, 38, 43, 46, 49, 51, 53, 73, 75, 77, 91, 92: Added ISBN and country when missing.
- Reference 78: Software reference has been modified to adapt MDPI guidelines.
We hope to have answered all your considerations. Again, thank you very much for your time and effort in reviewing our manuscript. Best regards.